# Longitudinal trends and determinants of patient-reported side effects on ART–a Swedish national registry study

Åsa Mellgren[1,2], Lars E. Eriksson[3,4,5], Maria Reinius[3], Gaetano Marrone[6,7], Veronica Svedhem[5,6]*

1 Department of Infectious Diseases, Sahlgrenska University Hospital, Gothenburg, Sweden, 2 Department of Infectious Diseases, Institute of Biomedicine, Sahlgrenska Academy, University of Gothenburg, Gothenburg, Sweden, 3 Department of Learning, Informatics, Management and Ethics, Karolinska Institutet, Stockholm, Sweden, 4 School of Health Sciences, City, University of London, London, United Kingdom, 5 Department of Infectious Diseases, Karolinska University Hospital, Stockholm, Sweden, 6 Unit of Infectious Diseases, Department of Medicine Huddinge, Karolinska Institutet, Stockholm, Sweden, 7 Department of Global Public Health, Karolinska Institutet, Stockholm, Sweden

* veronica.svedhem-johansson@sll.se

**Data Availability Statement:** All data files are available from the National assurance registry InfCareHIVdatabase from the day of accepted for

## Abstract

### Introduction

The use of patient-reported outcomes (PROs) to systematically quantify adverse events (AE) will assist in the improvement of medical care and the QoL of patients living with HIV (PLWH). The aim of this study was to investigate the associations between self-reported side effects and other PROs, demographics and laboratory data, and further evaluate the Health Questionnaire (HQ) as a tool for following trends in patient-reported side effects over time in relation to trends in prescribed third agent in ART.

### Materials and methods

The Swedish National Registry InfCareHiv includes an annual self-reported nine-item HQwhich is used in patient-centered HIV care in all Swedish HIV units. In this study, the experience of side effects was addressed. We analyzed 9,476 HQs completed by 4,186 PLWH together with details about their prescribed ART and relevant biomarkers collected during 2011–2017. Data were analyzed using descriptive statistics, Pearson's correlation coefficient and mixed logistic regression.

### Results

The cross-sectional analysis of the HQs showed that the frequency of reported side effects decreased from 32% (2011) to 15% (2017). During the same period, there was a shift in ART prescription from efavirenz (EFV) to dolutegravir (DTG) (positive correlation coefficient $r = 0.94$, p = 0.0016). Further, PLWH who reported being satisfied with their physical health (OR: 0.47, p = <0.001) or psychological health (OR: 0.70, p = 0.001) were less likely to report side effects than those less satisfied.

publication.(https://qrcstockholm.se/register/anslutna-register/infcarehiv-och-infcarehepatit/).

**Funding:** VS has received funding from the Gilead Sciences Nordic Fellowship Programme and the Department of Communicable Disease Control Health Protection Public Health Agency of Sweden. The funders had no role in study design, data collection and analysis, decision to publish, or preparation of the manuscript.

**Competing interests:** The authors have declared that no competing interests exist.

## Conclusions

Self-reported side effects were found to have a close relationship with the patient's ratings of their overall health situation and demonstrated a strong correlation with the sharp decline in use of EFV and rise in use of DTG, with reported side effects being halved. This study supports the feasibility of using the HQ as a tool for longitudinal follow up of trends in PROs.

## Background

The assessment of patient-reported outcomes (PROs) is emerging as an important tool to ensure the long-term health and improvement in quality of life (QoL) of people living with HIV (PLWH) [1]. The use of PROs to improve the treatment and care of PLWH has been found to be a valuable addition to clinicians' documentation of laboratory data and biomarkers related to adverse events (AE) [2]. The FDA, and others, recommend that the pharmaceutical industry should include PROs in the clinical outcome assessment (COA) of clinical trials to support the development of new drugs [3] and to let PROs decide treatment choice when studies show non-inferiority. It is well known that the reporting of adverse drug reactions (ADRs) by physicians and patients does not always correspond to clinical trial data and several factors may contribute to this discrepancy [4, 5]. In clinical trials, all unwanted medical events are registered as AEs or serious AEs regardless of whether they are linked to the treatment or not. Causality is difficult to determine since placebo-controlled studies are rare [6]. However, up to a quarter of patients who receive placebo drugs also report ADRs, a phenomenon often referred to as a nocebo effect [7], which is defined as a symptom perceived as negative in response to a sham treatment [8]. Several of the mechanisms known to underlie the nocebo effect, such as the patient-doctor relationship, negative affectivity, and experienced health, can also be detected by PROs [9, 10]. Managing and avoiding AEs and ADRs is important since they result in hospitalization [11, 12] and lead to poorer adherence to antiretroviral treatment (ART) [13, 14]. Observational studies are necessary to be able to unravel real-world data regarding the prevalence of present, slowly evolving and rare ADRs. Qualitative studies can gain a deeper understanding of patients' experiences of side-effects [15, 16].

There is a call for further research to demonstrate how the use of PROs in the field of HIV can provide a clear link between improvements in PROs and improvements in health and in clinical care [1]. The aim of this study was to investigate the associations between self-reported side effects and other PROs, demographics and laboratory data, and evaluate the HQ as a tool for following trends in patient-reported side effects over time in relation to trends in prescribed third agent in ART. We also conducted qualitative interviews with PLWH to gain a deeper understanding of what patients receiving HIV care include within the concept of side effects.

## Material and methods

### The InfCareHIV registry and the Health Questionnaire

InfCareHIV is a web-based, Swedish national quality assurance registry, established in 2008, with 100% coverage of PLWH in Sweden; it currently holds data on 11,064 patients. All HIV care in Sweden is performed within the public health service and microbiological and CD-4 data are transmitted directly from the analyzing laboratory to the InfCare registry. Data on ART is registered manually by an administrator, who is also a healthcare professional, based on information in the medical records. An automated quality check of incoming data takes

place annually with the help of the DDM-tool, an instrument created in collaboration with EuroCoord. The Swedish Association of Local Authorities and Regions awarded the InfCareHIV registry the highest rating for data quality of all National Quality Registers in Sweden [17]. PLWH are consecutively enrolled in the registry at time of HIV diagnosis, and ART and laboratory data are monitored at least every six months. The InfCareHIV registry also serves as a database and a clinical support tool [14].

In 2011, a self-reported Health Questionnaire (HQ) assessing PROs was electronically integrated into InfCareHIV to be answered annually by patients either via a website or using a computerized or paper version at the patient's outpatient clinic. As well as the Swedish version of the HQ, there is also a validated English version available which contributed 8% of all responses in the study. Illiterate patients or patients not fluent in Swedish or English can be offered assistance from an interpreter.

The HQ consists of nine items (S1 Appendix) and has been validated and described in Marrone et al. [14]. The primary aim of the HQ is to assess PROs concerning physical, psychological and sexual health, adherence, side effects and feeling of involvement in care. The results from the HQ form a basis for discussions at the patient's routine clinical follow-up visit. The use of the HQ provides a more holistic view of the patient's health and well-being and enhances person-centered HIV care, focusing the consultation on the patient's current needs. For patients on ART (Item4a) who have experienced side effects (Item 4b), the severity of side effects (Item 4c) is assessed using responses on a five-point Likert scale. In the present study, the term side effects is used to refer to unintended AEs demonstrated by negative or adverse symptoms or physical changes where causality may be possibly, although not necessarily, related to the prescribed ART (e.g. ADRs).

## Study population

The data in this study derive from 4,186 individuals ≥18 years who had responded to the HQ at least once during the period 2011 to 2017. Fifteen of these also provided data through individual qualitative interviews. During the data collection period, PLWH received ART according to the Swedish HIV treatment guidelines; these were updated in 2009 [18], 2013 [19] and 2016 [20].

The following variables from the InfCareHIV were used in the analysis: sex, age, route of transmission, country of origin, date of HIV diagnosis, plasma HIV RNA (cop/ml), CD4-cell count ($x10^6$/l) and ART. Responses to all the questions on the HQ were analyzed.

## Statistical methods

Mean and standard deviation or median and interquartile range, t-test or the Wilcoxon rank-sum (Mann-Whitney) test were used to summarize numerical variables and compare their values between two groups, frequencies and percentages. A chi-square test was used to summarize and compare categorical variables. Pearson's correlation coefficient was used to test the correlation between numerical variables. Svyset Stata prefix was used to take into account the effect of data being clustered within patients. The answers to the questions concerning physical, psychological and sexual satisfaction were dichotomized into not satisfied (corresponding to Likert-scale answers: very unsatisfied, unsatisfied, rather unsatisfied and rather satisfied) and satisfied (corresponding to Likert-scale answers: satisfied and very satisfied) in accordance with other Swedish studies using the Li-Sat scale [21]). The answers concerning feeling involved in care (Item 5) were dichotomized into yes (corresponding to Likert-scale answers: sometimes, always) or no (corresponding to: never, seldom) and the answers concerning satisfied with care (Item 6) were dichotomized into satisfied (corresponding to Likert Scale

answers: satisfied or very satisfied) and not satisfied (corresponding to: very unsatisfied, unsatisfied, rather unsatisfied and rather satisfied).

A mixed logistic regression model with side effects (yes/no) as outcome was used to assess the determinants of self-reported side effects among patients by testing the following variables: gender, mode of transmission, born in Sweden or not, CD4 nadir, (were fixed variables), years since start of treatment, age, years since diagnosis, CD4 at time of questionnaire, viral load<50 cop/mL at time of questionnaire, reported number of missed doses of HIV medication, and ratings for satisfaction with physical health, psychological health and sexual life, satisfaction with HIV healthcare, and feeling involved in their own care. A backward stepwise regression model was used with the significance level for removal from the model set at $p<0.20$. Complete case analysis was used for the mixed regression model. Odds Ratios (OR) and their 95% confidence interval (CI) were estimated. P-values $<0.05$ were considered statistically significant. STATA version 15 (StataCorp, College Station, Texas, USA) was used for the analysis.

## Qualitative interviews

Fifteen participants were interviewed by the third author (MR) using a semi-structured interview guide to gain a deeper understanding of the participants' experiences of side effects. Participants were asked if they had experienced side effects, if side effects had interfered with their everyday life and how they would explain the concept of "side effects" (for the complete interview guide, see S1 Interview guide). The participants were recruited by research nurses in a process of purposeful sampling; the participants had been living with HIV for over a year, were durably virally suppressed, and could speak Swedish and/or English.

Interpretive description [22], an approach suitable for exploring actual real-world questions, was used as the analytic framework. Transcripts from the audio-recorded interviews were read through several times by MR and all text concerning side effects was selected and imported into Nvivo 11. All lines of text were first labeled in a process of open coding with the aim of identifying patterns in the transcripts, and a broad-based coding scheme was set up with descriptions of reoccurring codes. This coding scheme was then altered in the coding process, enabling the same line of text to be labeled with several codes. The members of the author group discussed the coding and, in a process of grouping codes together, identified two overall themes that corresponded to the aim of investigating patients' experiences of side-effects: 1) Participants' interpretations and explanations of the meaning of the concept *side effects of antiretroviral therapy* and 2) Participants descriptions of experienced AEs that negatively affected their life and well-being.

## Ethics

The study was approved by the regional Ethics Committee in Gothenburg (Dnr 293–16 and Dnr 579–18). The procedures for the qualitative interviews were approved by the Regional Ethics Committee in Stockholm (#2013/335–32). Informed written consent was collected prior to all interviews.

## Results

Between 2011 and 2017 the National InfCareHIV registry included 7,489 PLWH who were receiving care; 4,186 of these were included in this study. The participants had responded to a total of 9,137 questionnaires during the time period for the study. The study population and reasons for exclusion are described in a flow chart (Fig 1).

Each individual patient included had, on average, responded to the HQ twice (median, range 1–8), 398 patients responded to the HQ twice during a calendar year. Data concerning

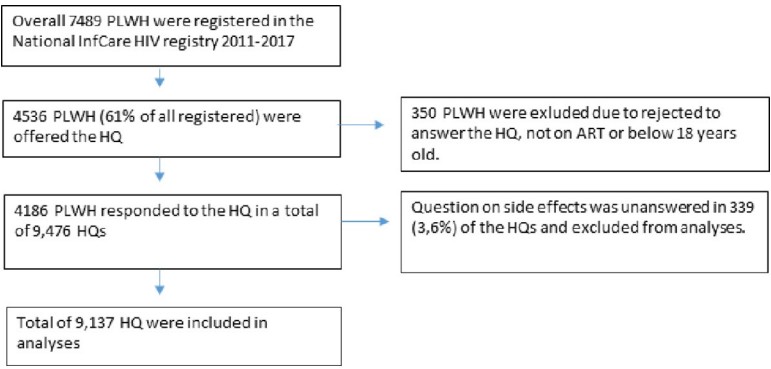

**Fig 1. Study population and number of Health Questionnaires performed.**

sociodemographics, epidemiology, CD4 cell count nadir, CD4 cell count and proportion of patients with an HIV viral load <50 cop/mL are presented in Tables 1 and 2 together with the results from the HQs including the patient-reported prevalence of side effects. Side effects were reported on 20.3% of the HQs (1,854 of 9,137). Of the patients who had experienced side effects, 1,808 (97.5%) specified the extent to which they were troubled by the side effects: 448 (24.8%) reported that they were affected to some or a large extent while 1,360 (75.2%) reported that they were affected to a minor or no extent. Fifteen PLWH, eight women and seven men, aged 30–64 years, participated in the qualitative interviews. All were Swedish citizens, although their country of origin varied, and they had been living with HIV for between 2 and 33 years.

## Frequency of patient-reported side effects by calendar year

During the study period, 1,301 (31.1%) PLWH reported experiencing side effects at least once. The cross-sectional analysis of the annual HQ data showed that the frequency of reported side effects decreased from 32% (196/620) in 2011 to 15% (197/1,330) in 2017 (Fig 2).

**Table 1. Patients' characteristics at their first visit (n = 4,186).**

| Characteristics | Total (Col %) |
| --- | --- |
| | 4,186 (100) |
| **Gender** | |
| Female | 1,493 (35.7) |
| Male | 2,693 (64.3) |
| **Age in years** (Mean, SD[@]) | 46.4 (11.5) |
| **HIV route of transmission** | |
| Heterosexual | 2,173 (52.2) |
| Men who have sex with men (MSM) | 1,480 (35.5) |
| Drug use | 247 (5.9) |
| Blood products | 58 (1.4) |
| Unknown/Other | 150 (3.6) |
| Missing | 19 (0.4) |
| **Country of origin** | |
| Abroad | 2,328 (55.6) |
| Sweden | 1,858 (44.4) |
| **CD4 cell count nadir in cells/mm3** (Median, Interquartile range) | 214 (150.1) |
| **Years since HIV diagnosis** (Mean, SD) | 10.8 (7.8) |
| Missing | 18 (0.4) |

**Table 2. Frequency of side effects reported by PLWH on ART in relation to gender, age, route of transmission, origin, CD4-cell count, missed doses of ART, PROs, time on ART and time since HIV diagnosis (9,137 HQs).**

| Characteristics | Side effects No (Row %) | Side effects Yes (Row %) | P-value |
|---|---|---|---|
| **HIV-related characteristics** | | | |
| **Total** | 7,283 (79.7) | 1,854 (20.3) | |
| **Gender** | | | 0.108* |
| Female | 2,553 (78.8) | 687 (21.2) | |
| Male | 4,730 (80.2) | 1,167 (19.8) | |
| **Age in years** (Mean, SD[@]) | 48.2 (11.4) | 47.7 (11.0) | 0.060** |
| **HIV route of transmission** | | | <0.001* |
| Heterosexual | 3,836 (80.2) | 945 (19.8) | |
| Men who have sex with men (MSM) | 2,477 (77.5) | 719 (22.5) | |
| Drug use | 563 (88.0) | 77 (12.0) | |
| Blood products | 91 (79.1) | 24 (20.9) | |
| Unknown/Other | 209 (76.6) | 64 (23.4) | |
| **Country of origin** | | | 0.827* |
| Abroad | 3,833 (79.6) | 981 (20.4) | |
| Sweden | 3,450 (79.8) | 873 (20.2) | |
| **CD4 cell count nadir in cells/mm3** (Median, Interquartile range) | 209.1 (142.7) | 202 (138.7) | 0.053*** |
| **CD4 cell count in cells/mm3 at HQ[&]** (Median, Interquartile range) | 600.4 (283.1) | 590.7 (259.7) | 0.18*** |
| **HIV Viral Load <50 cop/mL at HQ[&]** | | | 0.08** |
| No | 454 (76.4) | 140 (23.6) | |
| Yes | 6,613 (80.1) | 1,644 (19.9) | |
| **Years since start of ART[#]** (Mean, SD) | 9.5 (6.5) | 9.9 (6.8) | 0.007** |
| **Years since HIV diagnosis** (Mean, SD) | 12.2 (7.8) | 12.8 (8.2) | 0.002** |
| **Health Questionnaire results** | | | |
| **Missed ART[#] doses during the previous week** | | | <0.001* |
| 0 | 6,343 (80.8) | 1,511 (19.2) | |
| 1–2 | 752 (73.2) | 276 (26.8) | |
| 3+ | 101 (73.2) | 37 (26.8) | |
| **Satisfied with physical health** | | | <0.001* |
| No | 2,449 (71.6) | 973 (28.4) | |
| Yes | 4,798 (84.6) | 871 (15.4) | |
| **Satisfied with psychological health** | | | <0.001* |
| No | 2,540 (72.9) | 946 (27.1) | |
| Yes | 4,688 (84.1) | 889 (15.9) | |
| **Satisfied with sexual life** | | | <0.001* |
| No | 3,459 (75.8) | 1,104 (24.2) | |
| Yes | 3,403 (83.4) | 605 (16.6) | |
| **Satisfied with care** | | | <0.001* |
| No | 313 (69.3) | 139 (30.7) | |
| Yes | 6,906 (80.3) | 1,693 (19.7) | |
| **Feel involved in care** | | | <0.001* |
| No | 927 (71.3) | 373 (28.7) | |

(*Continued*)

**Table 2.** (Continued)

| Characteristics | Side effects No (Row %) | Side effects Yes (Row %) | P-value |
| --- | --- | --- | --- |
| Yes | 6,156 (81.2) | 1,422 (18.8) | |

@ SD = Standard Deviation

# ART = Antiretroviral Therapy

& HQ = Health Questionnaire

* Chi-squared test

** T-test

*** Wilcoxon rank-sum (Mann-Whitney) test.

Fig 3 shows the percentage per year of PLWH prescribed efavirenz (EFV) or dolutegravir (DTG), respectively, and percentage of PLWH reporting side effects. There was a statistically significant positive correlation coefficient between yearly percentage of patients prescribed EFV and percentage of patients reporting side effects ($r = 0.94$, $p = 0.0016$), and a statistically significant negative correlation between percentage of patients prescribed DTG and percentage of patients reporting side effects ($r = -0.83$, $p = 0.02$). The use of early generations of protease inhibitors (PI) decreased over the corresponding period; the use of lopinavir decreased from 7.5% to 1.2% and atazanavir (ATV) from 13.9% to 4.1%.

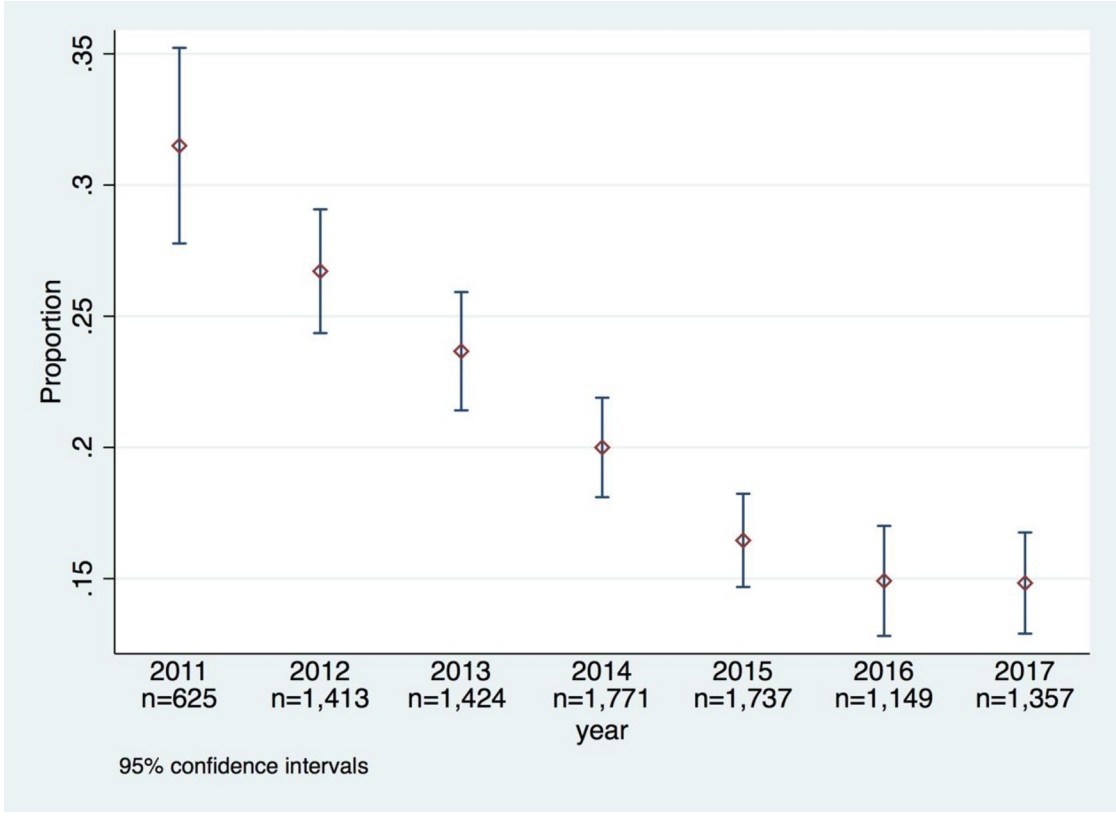

**Fig 2. Frequency of patient-reported side effects on the Health Questionnaire in the InfCareHIV Swedish national registry during 2011–2017.**

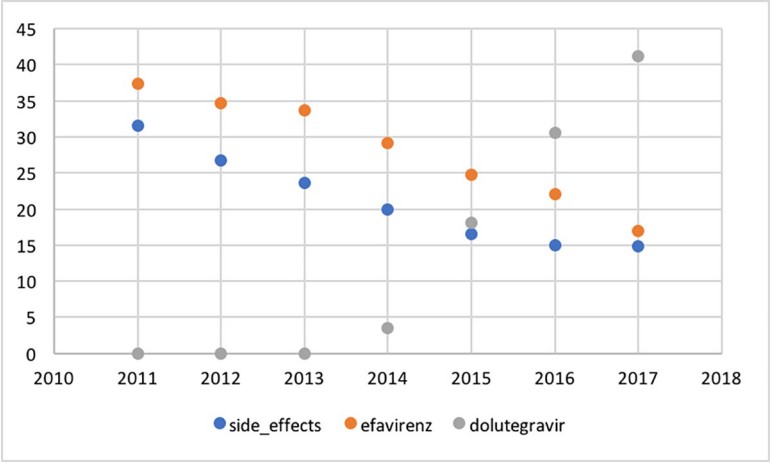

**Fig 3. Trend analysis of percentage per year of PLWH prescribed efavirenz or dolutegravir, respectively, and percentage of PLWH reporting side effects.**

## Multivariable analyses

The multivariable analyses, run on 7,835 observations (complete case analysis), showed that the reporting of side effects was statistically significantly associated with other PROs; however, the analysis does not determine causality. PLWH who reported being satisfied with their physical health (OR: 0.47, 95% CI 0.39–0.37), psychological health (OR: 0.70, 95% CI 0.57–0.84) or sexual life (OR: 0.74, 95% CI 0.62–0.89) were less likely to report side effects than those who were less satisfied. In addition, PLWH who felt involved in their care were less likely to experience side effects than those who did not feel involved in their care (OR 0.66, 95% CI 0.53–0.82). PLWH who reported having missed one or two doses of ART during the previous week were more likely to report side effects (OR: 1.50, 95% CI 1.17–1.91) compared to patients who reported no missed doses. There was a negative correlation between age and reporting of side effects and a borderline positive correlation between years since HIV diagnosis and CD4-cell count and reporting of side effects (Table 3).

## Associations between side effects and ART

The study cohort had received 9,331 prescriptions for a third agent in their ART (Table 4). Side effects were reported on 1,854 HQs, of which 1,831 patients were on ART including a third agent and 23 on ART without a third agent. Of those who reported side effects, the most common third agent was EFV followed by darunavir (DRV) and atazanavir (ATV) (Table 4). These three substances were also the most prescribed third agents (Table 4).

## Two themes were identified in the analysis of the qualitative interviews

The first theme was "Participants' interpretations and explanations of the meaning of the concept side effects of antiretroviral therapy". Most participants described side effects as being when the medication affected the body in a harmful way. One said that they occurred when the body resisted the medication and others that they only first considered a bodily reaction to be a side effect when their doctor had confirmed it. Others suggested that all events, both physical and social, that occurred while taking medication were side effects, since for example having to leave social activities early to go home to take your medication could be considered a side effect.

**Table 3. Logistic regression: Determinants for reporting having experienced side effects in 4,186 PLWH based on 9,137 Health Questionnaires from between 2011 and 2017.**

|  | Odds Ratio | 95% CI interval | P-value |
|---|---|---|---|
| **Age** | 0.98 | 0.97–0.99 | 0.001 |
| **Years since HIV diagnosis** | 1.01 | 1.00–1.03 | 0.051 |
| **CD4 cell count in cells/mm3 at HQ** | 1.00 | 1.00–1.00 | 0.020 |
| **Missed ART doses during the previous week** |  |  |  |
| 0 | ref | ref | Ref |
| 1–2 | 1.50 | 1.17–1.91 | 0.001 |
| 3+ | 1.03 | 0.54–1.95 | 0.937 |
| **Satisfied with physical health** |  |  |  |
| No | ref | ref | Ref |
| Yes | 0.47 | 0.39–0.57 | <0.001 |
| **Satisfied with psychological health** |  |  |  |
| No | ref | ref | ref |
| Yes | 0.70 | 0.57–0.84 | <0.001 |
| **Satisfied with sexual life** |  |  |  |
| No | ref | ref | ref |
| Yes | 0.74 | 0.62–0.89 | 0.002 |
| **Feel involved in care** |  |  |  |
| No | ref | ref | Ref |
| Yes | 0.66 | 0.53–0.82 | <0.001 |

**Table 4. Prevalence of prescribed third agent in PLWH experiencing side effects and of prescribed third agent in ART of PLWH who completed the Health Questionnaire during 2011–2017.**

| Third agent in ART of PLWH experiencing side effects | Number | Percentage |
|---|---|---|
| Efavirenz | 493 | 27.0 |
| Darunavir | 343 | 18.7 |
| Atazanavir | 317 | 17.3 |
| Dolutegravir | 202 | 11.0 |
| Raltegravir | 147 | 8.0 |
| Rilpivirine | 113 | 6.2 |
| Lopinavir | 108 | 5.9 |
| Nevirapin | 93 | 5.1 |
| Other | 15 | 0.8 |
| Total | 1831 | 100 |
| Third agent in ART of PLWH when completing the HQ | Number | Percentage |
| Efavirenz | 2,586 | 27.7 |
| Atazanavir | 1,529 | 16.4 |
| Darunavir | 1,478 | 15.8 |
| Dolutegravir | 1,270 | 13.6 |
| Rilpivirine | 772 | 8.3 |
| Raltegravir | 621 | 6.7 |
| Nevirapine | 517 | 5.5 |
| Lopinavir | 436 | 4.7 |
| Other | 122 | 1.3 |
| Total | 9,331 | 100 |

**Table 5. Possible adverse events and experiences that participants described as side effects.**

| Visible changes | Measurable changes | Non-measurable changes | Concerns about future SE | Effect of taking ART |
|---|---|---|---|---|
| Redistribution of body fat | Elevated liver enzymes | Altered sensation in the mouth | Accelerated aging | Difficulty swallowing tablets |
| Blood in urine | Elevated creatinine | Dizziness | Worrying about what is happening in their body | Nausea from thinking about the tablets |
| Brown urine | Neuropathy | Tiredness | | Limited social life |
| Thin legs | | Feeling awkward | | |
| Yellow eyes | | Joint pain | | |
| Yellow skin | | Nausea | | |
| Erectile dysfunction | | Pain | | |
| Having trouble walking | | Bloated stomach | | |
| Vomiting | | Feeling drunk | | |

The second theme was "Participants' descriptions of bodily sensations or experienced AEs that negatively affected their life and well-being". There was great variation in the type of bodily sensations and/or side effects described by the participants. These are summarized in Table 5 under the categories Visible changes, Measurable changes, Non-measurable changes, Concerns about future side effects (SE) and Effects of taking ART. Most participants said that they did not currently have side effects from their HIV medication. However, many participants reported having previously experienced side effects and some described bodily sensations or events, although they were not sure if they were connected to their medication. Participants repeatedly reported that it could be difficult to know if something they experienced was caused by their medication. Some also described concerns about the medicine possibly doing something to them that they were unaware of and which might only show signs when they were older. Several participants referred to the medicine as poison and expressed concerns that the medicine might cause internal damage, accelerate aging or have unknown long-term effects. Participants who had other chronic diseases as well as HIV described difficulty in knowing if experienced events, e.g. erectile dysfunction, were caused by one or other of the diseases or by the medication.

## Discussion

This observational study of 4,186 PLWH demonstrates that self-reported side effects are common and diverse, but that the reported prevalence had halved during the study period from 32% of the answered HQs in 2011 to 15% in 2017. The introduction of new antiretroviral agents probably contributed to this decrease since we found a correlation between the drop in reported side effects and alterations in the prescribed third agent in ART during the time period. The changes in ART prescription, recommended by the Swedish HIV treatment guidelines [18–20] and also presented in a previous study [23], were characterized by the decline in prescription of EFV and PI and the increased prescription of integrase inhibitors as third agents.

EFV has a well-known adverse reaction profile with neuropsychiatric side effects and the prescription of EFV has decreased from 25% in 2011 to 15.8% in 2017 [24]. DTG is well tolerated, has less organ impact and fewer interactions with other drugs, and a lower risk of discontinuation due to AEs compared to EFV [25]. Thus, we found a decrease in patient-reported side effects, a corresponding decrease in the prescription of drugs with well-known AEs and an increase in the prescription of DTG. In the future, when choosing between two ART with equivalent virologic outcome, PROs can be of major importance when making decisions

concerning treatment strategy, as is already the case in other medical fields such as oncology [26]. The HQ has proved to be a valid instrument for monitoring ART adherence [14] and it has now also been shown to be a useful tool for following trends in PROs that address side effects over time.

In the present study, PLWH who missed 1–2 doses of ART during the previous week were 1.5 times more likely to report experiencing side effects. The relationship between reported side effects and poorer adherence has been confirmed both in the Swedish population of PLWH [14] and internationally [27, 28] and interventions that support skills for managing side effects have demonstrated positive effects on ART adherence [29].

The analysis of the PROs on the HQs showed that reports of side effects were least prevalent among those who reported being satisfied or very satisfied with their physical health. In these patients the reports of side effects were lower by more than half compared to those who rated their physical health as less satisfactory. There was also an association with the individual's reported satisfaction with their psychological health, where the reports of side effects were about one-third lower among those reporting being satisfied with their psychological health compared to those who were less satisfied. This association between PROs, where high self-rated health was a determinant for fewer experienced side-effects, highlights the side-effect issue as a central topic that should be raised in the patient's interactions with healthcare professionals. Assessing the patient's experience of their overall global health gives the opportunity to take measures to support PLWH in the improvement of their quality of life.

The qualitative interviews with PLWH showed that, although virally suppressed, they described a variety of negative bodily sensations that they related to their treatment. The experiences of side effects included AEs that were visible and measurable as well as mental processes and experiences in the form of negatively charged thoughts where it was unclear if causality was related to the prescribed drugs. Some even suggested that a side effect could be any kind of negative effect associated with the ART, including impacts such as difficulty swallowing tablets or having to adjust their life to enable them to take their medication in private. These thoughts and experiences, including concerns about ongoing and expected future symptoms, some probably due to a lack of knowledge about the medical effects of the ART, are generally not counted as AEs in clinical trials and this could be built upon to further understand the nocebo concept.

When working with patients experiencing side effects, it could be worth acknowledging that it may be difficult for patients to understand whether or not a negative bodily sensation they are experiencing is linked to their ART or to a particular drug. Other aspects worth acknowledging are that patients may fear that their treatment will be harmful from a long-term perspective and that effects related to parallel diagnoses and treatments as well as residual side effects from previous ART can be difficult to distinguish from the effects originating from their current ART.

Side effects may be a reason for changing ART and studies have shown that this can improve the quality of life of PLWH [30]. Nevertheless, when a patient is experiencing side effects it is important to understand the underlying cause in order to be able to improve their QoL, which may not necessarily involve a change in ART.

In this study, PLWH who reported that they felt involved in their care were less likely to report side effects. Involvement in care may reduce the prevalence of side effects both as a result of the patient's own expectations being addressed and also through well-functioning patient-caregiver communication [31–33]. Additionally, since the PROs were assessed over time in the study, part of the decline in prevalence could be a result of an increase in patient-centered HIV care where the experience of side-effects was addressed.

Study constraints consist mainly of the fact that parameters that may be of importance for people's experiences of side effects, such as co-morbidity, level of education, work life situation, family situation, alcohol and drug use, are not available in InfCareHIV. We have not assessed the decline of side effects according to prescribed third agent at individual level but instead at an ecological level; the relationship between side effects and prescribed third agent at individual level could not be assessed since the HQ is performed annually without relation to ART changes. The HQ also does not capture the exact symptoms that the patients are experiencing; the HQ is designed to be brief and to facilitate the clinical consultation. We therefore performed the qualitative part of this study to investigate the aspects that patients include in the concept of side-effects, which was found to be a considerably diverse range of experienced side effects.

## Conclusion

In this study, reported side effects were found to have a close relationship with the patient's overall global health and a strong correlation with the sharp decline in the use of EFV and rise in the use of DTG, with reported side effects being halved. This study supports the feasibility of using the HQ as a tool for longitudinal follow up of trends in PROs.

## Supporting information

**S1 Appendix.**
(DOCX)

**S1 Interview guide.**
(DOCX)

## Acknowledgments

We thank Pernilla Albinsson, Department of Infectious Diseases, Karolinska University Hospital, for help with the Health Questionnaire.

## Author Contributions

**Conceptualization:** Veronica Svedhem.

**Data curation:** Gaetano Marrone, Veronica Svedhem.

**Formal analysis:** Maria Reinius, Gaetano Marrone.

**Funding acquisition:** Veronica Svedhem.

**Investigation:** Åsa Mellgren, Veronica Svedhem.

**Methodology:** Lars E. Eriksson, Maria Reinius.

**Project administration:** Veronica Svedhem.

**Resources:** Veronica Svedhem.

**Supervision:** Veronica Svedhem.

**Validation:** Lars E. Eriksson.

**Visualization:** Åsa Mellgren, Maria Reinius, Gaetano Marrone, Veronica Svedhem.

**Writing – original draft:** Åsa Mellgren, Veronica Svedhem.

**Writing – review & editing:** Åsa Mellgren, Lars E. Eriksson, Maria Reinius, Gaetano Marrone, Veronica Svedhem.

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
