## [Decision Letter · Decision Letter 0]

12 Aug 2020

PONE-D-20-19335

Longitudinal trends in patient-reported outcomes addressing side effects and prescribed third agent in ART – a Swedish national registry study

PLOS ONE

Dear Dr. Svedhem,

Thank you for submitting your manuscript to PLOS ONE. After careful consideration, we feel that it has merit but does not fully meet PLOS ONE’s publication criteria as it currently stands. Therefore, we invite you to submit a revised version of the manuscript that addresses the points raised during the review process.

You will see that the Referees found your work of some interest. However, they also raised major criticisms (see comments by Reviewer #2 and #3), and did not grant your paper enough priority to recommend publication in its form. However, if you think all objections raised by the referees can be considered and if additional data requested by reviewers can be provided, we may be willing to reconsider your manuscript.

We look forward to receiving your revised manuscript.

Kind regards,

Giuseppe Vittorio De Socio, MD, PhD

Academic Editor

PLOS ONE

Journal Requirements:

2. Please include your ethics statement in the online submission form.

3. Please include additional information regarding the interview guide used in the study and ensure that you have provided sufficient details that others could replicate the analyses. For instance, if you developed an interview guide as part of this study and it is not under a copyright more restrictive than CC-BY, please include a copy, in both the original language and English, as Supporting Information.

Reviewers' comments:

Reviewer's Responses to Questions

**Comments to the Author**

1. Is the manuscript technically sound, and do the data support the conclusions?

Reviewer #1: Yes

Reviewer #2: Yes

Reviewer #3: No

2. Has the statistical analysis been performed appropriately and rigorously? 

Reviewer #1: Yes

Reviewer #2: Yes

Reviewer #3: Yes

3. Have the authors made all data underlying the findings in their manuscript fully available?

Reviewer #1: Yes

Reviewer #2: Yes

Reviewer #3: Yes

4. Is the manuscript presented in an intelligible fashion and written in standard English?

Reviewer #1: Yes

Reviewer #2: Yes

Reviewer #3: Yes

5. Review Comments to the Author

Reviewer #1: Dear Authors,

your article is well written and cover the interesting topic of the use of PROs in clinical practice in PLWH. However, I suggest to revise the manuscript and the abstract for some grammar and punctuation errors before re-submission.

Reviewer #2: PLoS One

Longitudinal trends in patient-reported outcomes addressing side effects and prescribed third agent in ART - a Swedish national registry study - Mellgren Å. et al.

This longitudinal study aimed at investigating the associations between self-reported side effects in relation to other patient-reported outcomes (PROs), demographic and laboratory data, and prescribed third agent in ART (9,476 Health Questionnaires analyzed, from 4,186 people living with HIV).

PROs are a key tool for the implementation of patient-centered HIV care, and observational studies measuring PROs in a “real life” setting are needed to complement data from clinical trials. The study by Mellgren et al. offers a valuable contribution to the field.

Its major strengths are its longitudinal design with a large sample size and the use of mixed methods (quantitative survey + qualitative interviews) for data collection.

The manuscript is well written and pleasant to read.

However, several points need clarification or complementary analyses. I have also a series of minor comments for the authors to improve the presentation of their study.

Points which need clarification/complementary analyses:

MATERIAL AND METHODS

Study population:

There is a mismatch between the presentation of patients’ selection in the Study population section (“The data in this study derive from 4,186 individuals ≥ 18 years who had responded to the HQ at least once during 2011-2017”) and its presentation in the Results section (“the question regarding experience of side effects was unanswered in 3.6% of the HQs; these HQs were therefore excluded from the analyses”).

Was the selection made on patients or on visits?

The first sentence suggests that patients who responded at least once to the HQ (any item or items specifically related to side effects? This should be explained) were selected – so, a selection on patients.

But I also understood that all patients analyzed were receiving ART (i.e. a selection was first made on visits, excluding visits without treatment).

This should be better explained (perhaps with a flow chart?).

Comparison tests:

- It’s unusual to present comparisons of patients’ characteristics (Table 1) using tests based on repeated measures (even if cluster effect is taken into account using a Stata command). “Visits” are compared, and not individuals. This makes difficult the understanding of the study population’s characteristics. I suggest the authors present characteristics of patients at the time first HQ was completed.

- The authors used t-test to compare continuous characteristics between patients who reported side effects and those who did not. This test better suits to normally distributed variables, such as (probably) age and other “duration” variables (time since HIV diagnosis and time since ART initiation) but not to CD4 counts. I suggest using the Wilcoxon rank-sum (Mann-Whitney) test to compare nadir and CD4 counts between groups, and to present median [interquartile range] instead of mean (SD) in Table 1 for these two variables.

Graphical representation of change in the percentage of patients reporting side effects:

Figure 1 presents, for each year of the study period (2011-2017), the percentage of patients who reported side effects.

- I suggest not to connect points, as the sample of patients is not the same for each year (“each included individual had responded on average three times”).

- A 95% confidence interval should be added for each percentage in the figure.

Figure 2 is interesting and raises the following question: why not testing a time-dependent variable “Prescribed third agent: EFV, DTG, other” directly in the models? (I’m not convinced by the explanation given line 319 to 323).

Mixed-effects logistic regression models:

The authors analyzed factors associated with report of side effects. Previous studies in the literature showed that self-reported symptoms could impact both quality of life (QoL) and adherence to treatment. I think that testing QoL-related variables (i.e. satisfaction with physical health, psychological health, sexual life) and missed ART doses as potential correlates of side effects may pose a problem of reverse causality. The authors should remove these four variables from the models (maybe just show their correlations with side effects at first time HQ was completed – Table 1).

It’s unclear which variables were fixed and which were time-dependent in the models. I think gender, HIV route of transmission, country of origin, and CD4 count nadir were fixed, and the other variables were time-dependent – is this the case?

Administration mode of the HQ:

Is information about the administration mode of the HQ (paper, web, computerized…) available? Can it be tested in the model?

Literacy/foreign language:

Can the authors document the percentage of patients who needed assistance to fill in the HQ? (this could also be tested in the model).

Minor comments:

ABSTRACT

- Line 45: “efavirence” must be corrected into “efavirenz”.

- The conclusion of the abstract should contain a short sentence about the feasibility of using the HQ (as in the conclusion of the manuscript).

MATERIAL AND METHODS

Statistical methods:

- Please cite the complete reference of the statistical software used for the analysis of quantitative data.

RESULTS

Table 1:

- Measurement units should be added in the variable label (age in years, CD4 in cells/mm3).

- Please add information on missing values (for instance, adding the % of missing values between parentheses for each variable).

- Reformulations: “Route of transmission” should be replaced with “HIV route of transmission”, “CD4 nadir” with “CD4 cell count nadir”, “Missed doses previous week” with “Missed ART doses during the previous week”, “Viral load<50” with “HIV viral load<50”, “Years since diagnosis” with “Years since HIV diagnosis”.

- Please add the spell-out form of all abbreviations (ART, SD, HQ) as a footnote.

- “Socio-demographic characteristics of respondents” include gender, age, and country of origin. HIV route of transmission, CD4 nadir, HIV viral load, years since start of ART, years since diagnosis and CD4 cell count at HQ should be presented under a different subtitle (perhaps “HIV-related characteristics”).

- Please add information on the type of statistical tests used as a footnote.

DISCUSSION

The lack of information in the HQ about the type of symptoms perceived should be acknowledged as a limitation for the study.

Reviewer #3: General comment:

This is a study that aims to summarize patient-reported aniretroviral drug side effects in a large number of HIV infected persons from a Swedish national registry from 2011 to 2017. The data analysis is based on the 9-item Health Questionnaire (HQ) submitted to the registry annually. The main study finding (a correlation between side effects and usage of 3rd antiretroviral agent) is not based on an individual level analysis and represents the main limitation of the study.

The major difficulty encountered in the review is the reporting of results. Sometimes the number of patients is reported and at other times number of HQ responses. However, when a number is reported it is often not clear whether it refers to the number of individuals or number of HQ. Since the main study aim is to look at a 7-year trend I would also expect that the study population of each individual year would be described. The time trend is presently presented only in percentages (no absolute numbers) of a limited number of features (side effects, efavirenz and dolutegravir usage). If space is a concern this data could be presented as s supplement. There is also little data (except on 15 interviewed persons) on the type of side effects patients experienced.

Specific comments:

Abstract

Please omit the subheading “Discussion” as this is usually not part of an abstract. The conclusion may include implications of the study, but they should reflect the main findings. Please consider revising the concluding sentence. Since no counselling intervention has been done, I am not sure the conclusion is based on the findings of the study.

Efavirenz is also misspelled (“efavirence”).

Background

Line 61: Please check the wording “biometric”. Are those “biometrics” biomarkers or CD4 cell counts and HIV1-RNA tests or something else?

Line 82: The aim of the study includes only the third ART agent (see discussion).

Material and Methods

Would suggest providing more information on the InfCareHIV registry. For example, how is the laboratory and ART data collected? Is it thru labs and pharmacies or they depend on entry by health care professionals or something else?

Please also report how the questions that have a Likert scale are reported? For example, in the Result section Table 1 reports on items from the HQ as yes/no whereas in the questionnaire many items (How satisfied are you with your physical health, How satisfied are you with your psychological wellbeing, How satisfied are you with your sexual life, Do you feel involved in the planning and realization of your HIV care and treatment? How satisfied are you with the quality of care provided at your HIV clinic) have a 4, 5 or 6 point scale. So, please indicate which answers were considered “yes” and which “no”.

Study population

Please also explain why only the “third agent” of an ART regiment was analyzed. Side effect can also occur from the NUCs backbone. What about patients on dual therapy or those who used more than 3 drugs? Were they excluded? This should be reported in the Method section.

Statistical methods:

Line 116. The chi-square test is used for categorical variables, so please consider rephrasing the sentence.

Please state which was the outcome variable in mixed logistic regression. The phrase “ratings of side effects” might suggest that the answer 4c of the HQ was used, although I would assume that the answer to 4b was used. How many patients were included into the multivariable model? From Table 1 it seems that there were some missing data (individual responses do not add to total responses) so this needs to be mentioned. It seems that the multivariable model did not include the type of ART given to the patients despite the fact that the relationship of 3rd agent and self-reported side effects was the main study aim. Inclusion of ART type might give a direct association of ART type and side effects. Was there a specific reason why ART was not included into the multivariable model? If so, this should be explained. Was calendar year of assessment included into the model? Was to model checked for multicollinearity? Was the model checked for significant interactions?

Results:

Please clarify the numbers presented in the first paragraph. If 4186 PLWH filled out a total of 9476 questionnaires then, on average, one person filled out 2.3 HQ. Yet, the last sentence of the paragraph suggests that the average was 3. Also, when referring to percentages from a sample please include the numbers from which these percentages are derived from. For example, if 562 is 5.6% then 100% should be 10036. But this figure is not mentioned previously. Also, it is not clear what is the absolute number of those who did not answer the question regarding experience of side effect (3.6% of what number?). I would suggest making a flow chart clearly outlining the study population and the number of HQs analyzed. The HQ was filled out once a year over a period of 7 years (see Method section), yet the range of individual responses was from 1 to 8 (line 159). This means that at least one patient had filled 2 HQ in a calendar year. Please clarify.

Line 160: What does the “biomarker data” refer to? If it is the CD4 cell count and viral load this could be mentioned instead of naming them biomarkers. What was the total number of patients that had self-reported side effects? The Results sections reports mainly on the number of HQ and not on number of patients.

Table 1. needs to be revised. It is presently a mix of frequencies based on the number of patients and frequencies based on the number of questionnaires filled out. This is confusing and very difficult to follow. Tables should be self-explanatory (without reading the text). All numbers should be reevaluated or explained. For example, it is not clear how many females had side effects. The total number of females is 1493, yet Table 1 reports 2553 with no side effects (probably number of HQ, not no. females) and 687 did have side effects (number of HQ?). Vertical sums in columns 2 and 3 also do not add to the totals on top of table (missing data?). The same is true for many other numbers (but not for all). It is also difficult to understand the meaning of means for some variables (age, CD4 cell count). For example, the “total” mean for age is lower than the mean in those with and without side effects. This should also be explained. It seems that one patient contributed to the mean age several times. To avoid misunderstandings, would suggest separate tables on number of patients and number of responses or some other clear presentation of data. Also, some characteristics on the bottom of Table 1 (viral load, years since ART and diagnosis and CD4 cell count at HQ) are under the subheading “Health questionnaires results” could be moved to another or a new subheading.

The annual HQ data is reported only in percentages, please add the absolute annual number. Please consider moving the sentence starting on line 174 (During the same period….) elsewhere (is not a result of the study).

In text, there are two figures referred to Fig.2. One in line 180 states that “Fig. 2 shows the percentage….“. The other at line 188 states „Fig2“. Logistic regression:……“. Please clarify.

For Fig 3.tiff referred in text as Fig.2. could you also report the absolute numbers (not only percentages). The vertical (y) and horizontal (x) axis have no titles. Any data on the nucleoside backbone usage? Please add the letter for the correlation coefficient before 0.94 and -0.83). This figure actually repeats the line for fig1.tiff. So, there is no need to have both Fig1.tiff and Fig3.tiff.

As some variables were omitted from multivariable logistic regression analysis, please provide results of the bivariable analysis in a supplemental table. Fig 2.tiff is incomplete. Figures should also be self-explanatory. No reference categories are included for categorical variables (yes vs no or something else). No units of measurements for age (per 1 year?, per 10-year?), CD4-cell count (per 1 cell? Per 10 cells?), year since HIV-diagnosis (per 1-year?) are mentioned. “Psyche” is jargon. It is difficult to see from Fig. 2.tiff the “negative correlation” between age and side effects, and “positive” for years of diagnosis and CD4 cell count (see lines 199-201). This might be because a small unit of measurement has been used for those continuous variables. Consider increasing the unit. However, I do assume that “(Fig 3)” mentioned on line 201 is the figure 2.tiff. If so, please correct the reference to figure. It would be useful to readers to have a footnote explaining what an OR < 1 or an OR > 1 means. I suspect that the line range are 95% CI, but this should be mentioned in a footnote.

Fig 3. Lines 202 to 204 is also mentioned as fig 2 on line 180. In the section on “Associations between side effects and ART” it is not clear how many individuals received prescriptions. The total number of prescriptions is presented (n=9311), but are those also the number of patients? I would assume that 9331 prescriptions have been recorded at the time of HQ reporting, but this needs to be clear in text (can be done in the Method section). Tables 2 and 3 can be combined. There is no need to report the cumulative percentage, and the decimal place can be set to 1. There is no mention of results obtained from question 4c (To what extent are you troubled by medical side effects?). Since the focus of the paper is on side effects it would be useful to have this data.

Consider rephrasing the title of Table 4. into “Possible adverse drug events and…….”

Discussion:

Consider expanding the limitations of the study (not able to make a cause-effect conclusion, no type of side effects recorded….).

6. PLOS authors have the option to publish the peer review history of their article (what does this mean?). If published, this will include your full peer review and any attached files.

Reviewer #1: No

Reviewer #2: **Yes: **Fabienne MARCELLIN

Reviewer #3: No

---

## [Author Response · Author response to Decision Letter 0]

6 Nov 2020

Reviewer # 1: 

Dear Authors, your article is well written and cover the interesting topic of the use of PROs in clinical practice in PLWH. However, I suggest to revise the manuscript and the abstract for some grammar and punctuation errors before re-submission

Author reply: We thank reviewer 1 for these highly supportive comments. We have made a thorough review of the text and revised it to improve the language.

Reviewer # 2

2.1.1. There is a mismatch between the presentation of patients’ selection in the Study population section (“The data in this study derive from 4,186 individuals ≥ 18 years who had responded to the HQ at least once during 2011-2017”) and its presentation in the Results section (“the question regarding experience of side effects was unanswered in 3.6% of the HQs; these HQs were therefore excluded from the analyses”).

The first sentence suggests that patients who responded at least once to the HQ (any item or items specifically related to side effects? This should be explained) were selected – so, a selection on patients. But I also understood that all patients analyzed were receiving ART (i.e. a selection was first made on visits, excluding visits without treatment). This should be better explained (perhaps with a flow chart?).

Author reply: Thank you for this suggestion. We have now made a flow chart (new figure 1) to clarify the study population and corrected the text in line 196-200. 

2:2:1 It’s unusual to present comparisons of patients’ characteristics (Table 1) using tests based on repeated measures (even if cluster effect is taken into account using a Stata command). “Visits” are compared, and not individuals. This makes difficult the understanding of the study population’s characteristics. I suggest the authors present characteristics of patients at the time first HQ was completed.

Author reply: Following the reviewer´s comments, we have now divided Table 1 into Table 1a and Table 1b: in Table 1a we present the Patients’ characteristics at their first visit (n=4,186) while in Table 1b we present the data from the Health Questionnaires. 

2:2:2 The authors used t-test to compare continuous characteristics between patients who reported side effects and those who did not. This test better suits to normally distributed variables, such as (probably) age and other “duration” variables (time since HIV diagnosis and time since ART initiation) but not to CD4 counts. I suggest using the Wilcoxon rank-sum (Mann-Whitney) test to compare nadir and CD4 counts between groups, and to present median [interquartile range] instead of mean (SD) in Table 1 for these two variables.

Author reply: We have followed the reviewer´s advice and have modified the hypothesis tests accordingly in Table 1b.

2:3:1 Figure 1 presents, for each year of the study period (2011-2017), the percentage of patients who reported side effects.

2:3:1a- I suggest not to connect points, as the sample of patients is not the same for each year (“each included individual had responded on average three times”).

-2:3:1b A 95% confidence interval should be added for each percentage in the figure.

Author reply: Thank you for your advice. We have made a new figure (Figure 2) 

2:3:2 Figure 2 is interesting and raises the following question: why not testing a time-dependent variable “Prescribed third agent: EFV, DTG, other” directly in the models? (I’m not convinced by the explanation given line 319 to 323, 

Author reply: The Health Questionnaire (HQ) data are captured annually without specification of treatment change. We don’t have systematically collected data with the aim of evaluating PROs with respect to each ART. The interesting analysis you suggest requests a completely different design with data collection to follow a protocol for monitoring substance change, treatment outcome and patient evaluation of adverse event (AE) after changing 1-3 substances, preferably with a prospective design, similar to a clinical trial. Therefore, the analyses presented in the manuscript represent those made at an ecological correlation, not individual level data. In the present study we wanted to focus on our initial aim: to evaluate the HQ as a tool for following trends in patient-reported side effects over time in relation to the trends in prescribed third agent. To clarify this, we have changed the text in study limitations, row 433-434.  

2:4:1 The authors analyzed factors associated with report of side effects. Previous studies in the literature showed that self-reported symptoms could impact both quality of life (QoL) and adherence to treatment. I think that testing QoL-related variables (i.e. satisfaction with physical health, psychological health, sexual life) and missed ART doses as potential correlates of side effects may pose a problem of reverse causality. The authors should remove these four variables from the models (maybe just show their correlations with side effects at first time HQ was completed – Table 1).

Author reply: In earlier work, as well as in the present study, we have chosen to analyze and present associations and relationships between PROs rather than causality (since QoL is dependent on several factors). We have clarified this in row 296-297. Table 1b now provides the requested correlations. 

2:4:2 It’s unclear which variables were fixed and which were time-dependent in the models. I think gender, HIV route of transmission, country of origin, and CD4 count nadir were fixed, and the other variables were time-dependent – is this the case?

Author reply: Yes, the reviewer is correct, we have clarified this in line 153.

2:5 Administration mode of the HQ:

2:5:1 Is information about the administration mode of the HQ (paper, web, computerized…) available? Literacy/foreign language. Can it be tested in the model? Can the authors document the percentage of patients who needed assistance to fill in the HQ? (this could also be tested in the model).

Author reply: The Health Questionnaire cohort includes patients from 220 nations and language issues are a common source of difficulties in communication in daily care, even if translators are frequently used. However, information about whether the HQ is completed by the patient on their own or with assistance from a healthcare professional or interpreter is not documented. Apart from the Swedish version, a validated English version exists and the English version was used by 8% of the cohort. Since the model includes country of birth, this variable would have a collinearity problem and therefore not be included in the model. The possibility for patients to answer a web version of the HQ was started in 2017 and during the first year only 75 (< 1%) patients chose this mode of response, we therefore judge that there are too few to have a potential effect if included in the present analysis, but we will include this as a possible covariate in analyses of future cohorts. We have added information about this in lines 105-110.

2:6 ABSTRACT

-2:6:1 Line 45: “efavirence” must be corrected into “efavirenz”.- The conclusion of the abstract should contain a short sentence about the feasibility of using the HQ (as in the conclusion of the manuscript). 

Author reply: We thank the reviewer for noting this and have corrected the text accordingly and added a sentence regarding feasibility in the abstract conclusions. 

2:7.1 Statistical methods:

- Please cite the complete reference of the statistical software used for the analysis of quantitative data.

Author reply: We thank the reviewer for reminding us about it. We have now added the full reference at the end of the “Statistical Methods” paragraph (lines 162-163).

2:8:1 Measurement units should be added in the variable label (age in years, CD4 in cells/mm3). Please add information on missing values (for instance, adding the % of missing values between parentheses for each variable). Reformulations: “Route of transmission” should be replaced with “HIV route of transmission”, “CD4 nadir” with “CD4 cell count nadir”, “Missed doses previous week” with “Missed ART doses during the previous week”, “Viral load<50” with “HIV viral load<50”, “Years since diagnosis” with “Years since HIV diagnosis”. Please add the spell-out form of all abbreviations (ART, SD, HQ) as a footnote. Socio-demographic characteristics of respondents” include gender, age, and country of origin. HIV route of transmission, CD4 nadir, HIV viral load, years since start of ART, years since diagnosis and CD4 cell count at HQ should be presented under a different subtitle (perhaps “HIV-related characteristics”). Please add information on the type of statistical tests used as a footnote.

Author reply: We thank the reviewer for this advice, which we have fully followed and have modified the text in Tables 1a and 1b.

2:9:1 The lack of information in the HQ about the type of symptoms perceived should be acknowledged as a limitation for the study.

Author reply: Thank you for this comment. We have added a sentence in line 435 in the limitations section. However, we find the qualitative interviews to be a strength in being able to obtain background facts on which to build further data collection concerning symptoms. 

Reviewer 3;1 The main study finding (a correlation between side effects and usage of 3rd antiretroviral agent) is not based on an individual level analysis and represents the main limitation of the study.

Author reply: We thank the reviewer for this very relevant comment. We agree that the complete picture of patients’ side effects would involve assessing the third agent used at an individual level. Since the HQ data are captured annually without specification to treatment change we don’t have systematically collected data with the aim of evaluating PROs with respect to each ART and correlations have therefore been made at an ecological level. For the present study, however, we wanted to focus on our initial aim: to evaluate the HQ as a tool for following trends in patient-reported side effects over time in relation to the trends in prescribed third agents. To clarify this, we have made a small change in the title and the aim in lines 34-37 and 84-87.

3:2 The major difficulty encountered in the review is the reporting of results. Sometimes the number of patients is reported and at other times number of HQ responses. However, when a number is reported it is often not clear whether it refers to the number of individuals or number of HQ. Since the main study aim is to look at a 7-year trend I would also expect that the study population of each individual year would be described. The time trend is presently presented only in percentages (no absolute numbers) of a limited number of features (side effects, efavirenz and dolutegravir usage). There is also little data (except on 15 interviewed persons) on the type of side effects patients experienced

Author reply: Thank you for these comments. We have revised the manuscript accordingly: the study population and number of HQ are presented in a flow chart in Figure 1, annual HQ in Figure 2, and we have added the study population in absolute numbers at the beginning and end of the study period in line 275. We have added to what extent the patients were troubled by the side effects in lines 208-211. We have also added the limitation regarding the type of side effects that the patients experience in lines 434-439. 

3:3 Abstract

Please omit the subheading “Discussion” as this is usually not part of an abstract. The conclusion may include implications of the study, but they should reflect the main findings. Please consider revising the concluding sentence. Since no counselling intervention has been done, I am not sure the conclusion is based on the findings of the study. Efavirenz is also misspelled (“efavirence”).

Author reply: We have revised the abstract and corrected the spelling. However, since some of the experiences were fears or other non-medical symptoms, we interpret the results as being that the counselling has contributed to the decline in side effects. All the HQs are followed up in routine clinical follow up visits. This has been clarified in line 113-116.

3:4 Background

3,4a: Line 61: Please check the wording “biometric”. Are those “biometrics” biomarkers or CD4 cell counts and HIV1-RNA tests or something else? 

Author reply: We thank the reviewer for this observation, we have corrected the sentence using the word biomarkers in line 64.

3.4b Line 82: The aim of the study includes only the third ART agent (see discussion).

Author reply: Yes, the aim of this study was to investigate the associations between self-reported side effects and other PROs, demographics and laboratory data, and further evaluate the Health Questionnaire (HQ) as a tool for following trends in patient-reported side effects over time in relation to trends in prescribed third agent in ART. 

3:5 Material and Methods

Would suggest providing more information on the InfCareHIV registry. For example, how is the laboratory and ART data collected? Is it thru labs and pharmacies or they depend on entry by health care professionals or something else?

 Author reply: The InfCareHIV registry has earned the highest rating for data quality of all National Quality Registers in Sweden (for reference please see http://kvalitetsregister.se/englishpages.2040.html. We have included additional information on the InfCareHIV registry in the Method in lines 94-102 to clarify this further.

3:5:3 Please also report how the questions that have a Likert scale are reported? For example, in the Result section Table 1 reports on items from the HQ as yes/no whereas in the questionnaire many items (How satisfied are you with your physical health, how satisfied are you with your psychological wellbeing, How satisfied are you with your sexual life, Do you feel involved in the planning and realization of your HIV care and treatment? How satisfied are you with the quality of care provided at your HIV clinic) have a 4, 5 or 6 point scale. So, please indicate which answers were considered “yes” and which “no”.

Author reply: We have added information about the dichotomization in the methods section in lines 141-150.

3:6 Study population

Please also explain why only the “third agent” of an ART regiment was analyzed. Side effect can also occur from the NUCs backbone. 

Author reply: We thank the reviewer for this comment, the answer is closely related to your comment 3:7:6-7. During the study time most of the changes made in the Swedish National HIV treatment Guidelines concerned the third agent, new substances were developed in NNRTIs and INIs, while we used the same Nucleosides. During recent years, more interest has been paid to the backbone and it is certainly true that NUCs also cause AE. However, the Health Questionnaire (HQ) data are captured annually without specification to treatment change. Thus, the data may be captured before or after treatment change, so relationships between side effects and prescribed third agent or backbone could not be assessed at an individual level. For this study we wanted to focus on our initial aim, to evaluate the HQ as a tool for following trends in patient-reported side effects over time in relation to the trends in prescribed third agents. This study shows that patients put into the concept of side effects more aspects than are usually considered as being AE in clinical trials and patients’ reports of side effects correlate with the decrease in prescription of efavirenz and increase in prescription of dolutegravir. A clinical trial with a protocol relating PROs to treatment change for each of the included components would be far beyond the aim of this study. The HQ would not be sufficient in that situation. We have expanded on this in the limitations section in line 434. 

3:6 b

What about patients on dual therapy or those who used more than 3 drugs? 

Were they excluded? This should be reported in the Method section

Author reply: We included all patients with a prescribed third agent and did not exclude those who used more than 3 drugs. The use of a third agent is presented in Table 3. 

3:7:1 Line 116. The chi-square test is used for categorical variables, so please consider rephrasing the sentence.

3:7:2 Please state which was the outcome variable in mixed logistic regression. 

3:7:3 The phrase “ratings of side effects” might suggest that the answer 4c of the HQ was used, although I would assume that the answer to 4b was used. 

3:7.4 How many patients were included into the multivariable model? 

3:7:5 From Table 1 it seems that there were some missing data (individual responses do not add to total responses) so this needs to be mentioned.

Author reply: We thank the reviewer for this advice. We have made changes accordingly on lines 135-138, 151, 297 and Table 1b. Ratings of side effects is Item 4c. 

3:7:6-7 It seems that the multivariable model did not include the type of ART given to the patients despite the fact that the relationship of 3rd agent and self-reported side effects was the main study aim. Inclusion of ART type might give a direct association of ART type and side effects. Was there a specific reason why ART was not included into the multivariable model? 

Author reply: We thank the reviewer for this very relevant comment that relates to comments 3:1 and 3:6 a. The Health Questionnaire (HQ) data are captured annually without specification of treatment change and therefore the data could not be assessed at an individual level. We have modified the title, the aim (34-37 and 84-87) and clarified that ART was not in the statistical model. 

3:7:8 Was calendar year of assessment included into the model? 

3:7:9 Was to model checked for multicollinearity? 

3:7:10 Was the model checked for significant interactions?

Author reply: We are grateful for the energy and time that the reviewer has given to our manuscript. The calendar year was not included in the model. The model was checked for collinearity using variance inflation factor (VIF) but none was found. The model was not checked for significant interactions for two reasons: firstly, the authors did not believe there was any particular interaction to be tested according to current knowledge about this topic, secondly, the model with interactions would have added unnecessary complexity to the interpretation of the parameters. 

Reviewer 3:8 Results:

3:8:1 Please clarify the numbers presented in the first paragraph. If 4186 PLWH filled out a total of 9476 questionnaires then, on average, one person filled out 2.3 HQ. Yet, the last sentence of the paragraph suggests that the average was 3. Also, when referring to percentages from a sample please include the numbers from which these percentages are derived from. For example, if 562 is 5.6% then 100% should be 10036. But this figure is not mentioned previously. Also, it is not clear what is the absolute number of those who did not answer the question regarding experience of side effect (3.6% of what number?). I would suggest making a flow chart clearly outlining the study population and the number of HQs analyzed.

Author reply: We understand that the presentation of the patient´s selection was unclear and we have now made a flow chart to clarify the study population (Figure 1). 

3:8:1c The HQ was filled out once a year over a period of 7 years (see Method section), yet the range of individual responses was from 1 to 8 (line 159). This means that at least one patient had filled 2 HQ in a calendar year. Please clarify.

Author reply: The patients were invited to respond to the web-based HQ two months prior to their annual follow-up appointment. Also, the patients’ regular visits sometimes occur within 12 months due to factors such as re-scheduled visits or holidays. This has resulted in several patients (n=398) responding to the HQ twice during a calendar year, we have inserted a comment about this in line 203. 

3:8:2 Line 160: What does the “biomarker data” refer to? If it is the CD4 cell count and viral load this could be mentioned instead of naming them biomarkers.

Author reply: We thank the reviewer for her/his patient consideration with the intention of increasing the readability of the text. We have now corrected the wording (now line 205).

3:8:2b What was the total number of patients that had self-reported side effects? The Results sections reports mainly on the number of HQ and not on number of patients.

Author reply: 1,301 patients (31.1%) experienced side effects at least once. This has been added to the manuscript (line 208).

3:8:3 Table 1. needs to be revised. It is presently a mix of frequencies based on the number of patients and frequencies based on the number of questionnaires filled out. This is confusing and very difficult to follow. Tables should be self-explanatory (without reading the text). All numbers should be reevaluated or explained. For example, it is not clear how many females had side effects. The total number of females is 1493, yet Table 1 reports 2553 with no side effects (probably number of HQ, not no. females) and 687 did have side effects (number of HQ?). Vertical sums in columns 2 and 3 also do not add to the totals on top of table (missing data?). The same is true for many other numbers (but not for all). It is also difficult to understand the meaning of means for some variables (age, CD4 cell count). For example, the “total” mean for age is lower than the mean in those with and without side effects. This should also be explained. It seems that one patient contributed to the mean age several times. To avoid misunderstandings, would suggest separate tables on number of patients and number of responses or some other clear presentation of data. Also, some characteristics on the bottom of Table 1 (viral load, years since ART and diagnosis and CD4 cell count at HQ) are under the subheading “Health questionnaires results” could be moved to another or a new subheading

Author reply: Table 1 has been revised and it has been divided into two sub-tables, Table 1a and Table 1b. We hope that the numbers are now clearer.

3:8:4 The annual HQ data is reported only in percentages, please add the absolute annual number.

Author reply: We have added the annual HQ in Figure 2. 

3:8:4 b Please consider moving the sentence starting on line 174 (During the same period….) elsewhere (is not a result of the study).

Author reply: We agree and have removed this sentence. 

Reviewer 3.8.5 a+b) In text, there are two figures referred to Fig.2. One in line 180 states that “Fig. 2 shows the percentage….“. The other at line 188 states „Fig2“. Logistic regression:……“. Fig 3.tiff referred in text as Fig.2 Please clarify.

Author reply: We have corrected the manuscript and revised the figures. 

3.8.5 c. Any data on the nucleoside backbone usage. 

Author reply: We thank the reviewer for this comment which is related to comment 3:6 a. Our intention was never to analyse side effects in relation to individual substances in ART.

3.8.6 a)could you also report the absolute numbers (not only percentages). 3.8.6 b)The vertical (y) and horizontal (x) axis have no titles

Author reply: Figure 2 has now been revised. 

3:8:7 Please add the letter for the correlation coefficient before 0.94 and -0.83

Author reply: We have corrected the manuscript and abstract accordingly. 

3.8.8 This figure actually repeats the line for fig1.tiff. So, there is no need to have both Fig1.tiff and Fig3.tiff.

Author reply: It is indeed correct that the trend line for Side-effects is expressed in both Figure 1 and Figure 3 and that the information is partly redundant. We have now updated “old” Figure 1 and added more information, it is now the new” Figure 2.

3.9,1 As some variables were omitted from multivariable logistic regression analysis, please provide results of the bivariable analysis in a supplemental table. 

Author reply: We thank the reviewer for this comment; however, we would prefer to better explain in the methods section, among the already mentioned variables tested in the model, the criteria for the selection of the variables in the final model. We have therefore added to the methods paragraph “A backward stepwise regression model was used with the significance level for removal from the model set at p<0.20” (lines 158-159).

3.9.2 a. 

3.9.2 Fig 2.tiff is incomplete. Figures should also be self-explanatory. No reference categories are included for categorical variables (yes vs no or something else). No units of measurements for age (per 1 year?, per 10-year?), CD4-cell count (per 1 cell? Per 10 cells?), year since HIV-diagnosis (per 1-year?) are mentioned. “Psyche” is jargon. It is difficult to see from Fig. 2.tiff the “negative correlation” between age and side effects, and “positive” for years of diagnosis and CD4 cell count (see lines 199-201). This might be because a small unit of measurement has been used for those continuous variables

Author reply: Following the relevant comments by the reviewer, we have chosen to show the results of the multivariable model in a table (Table 2) rather than in a figure. 

3.9.2b. Consider increasing the unit. However, I do assume that “(Fig 3)” mentioned on line 201 is the figure 2.tiff. If so, please correct the reference to figure. 

Author reply: Yes thank you, this fault has now been corrected.

3.9,2c It would be useful to readers to have a footnote explaining what an OR < 1 or an OR > 1 means. I suspect that the line range are 95% CI, but this should be mentioned in a footnote.

Author reply: We believe that these relevant comments by the reviewer are met by having Table 2 instead of the Figure. The interpretations of OR<1 or OR>1 are explained in the text. 

Fig 3. Lines 202 to 204 is also mentioned as fig 2 on line 180. In the section on “Associations between side effects and ART” it is not clear how many individuals received prescriptions. The total number of prescriptions is presented (n=9311), but are those also the number of patients? I would assume that 9331 prescriptions have been recorded at the time of HQ reporting, but this needs to be clear in text (can be done in the Method section). Tables 2 and 3 can be combined. There is no need to report the cumulative percentage, and the decimal place can be set to 1. There is no mention of results obtained from question 4c (To what extent are you troubled by medical side effects?). Since the focus of the paper is on side effects it would be useful to have this data.

Author reply: Thank you for the comments. We have revised the tables according to suggestions from you and the other reviewers. We have added descriptive statistics of the extent to which the patients were affected by side effects (lines 208-211). The number of prescriptions is not the same as patients, and we believe that the figure text is now sufficient. 

Consider rephrasing the title of Table 4. into “Possible adverse drug events and…….”

Author reply: We have rephrased the title according to the suggestion. 

3.9.4 Discussion:

Consider expanding the limitations of the study (not able to make a cause-effect conclusion, no type of side effects recorded….).

Author reply: As suggested, we have now expanded the Limitations.

---

## [Editor Report · Decision Letter 1]

9 Nov 2020

Longitudinal trends and determinants of patient-reported side effects on ART – a Swedish national registry study

PONE-D-20-19335R1

Dear Dr. Svedhem,

We’re pleased to inform you that your manuscript has been judged scientifically suitable for publication and will be formally accepted for publication once it meets all outstanding technical requirements.

Kind regards,

Giuseppe Vittorio De Socio, MD, PhD

Academic Editor

PLOS ONE
---

## [Editor Report · Acceptance letter]

24 Nov 2020

PONE-D-20-19335R1 

Longitudinal trends and determinants of patient-reported side effects on ART – a Swedish national registry study 

Dear Dr. Svedhem:

I'm pleased to inform you that your manuscript has been deemed suitable for publication in PLOS ONE. Congratulations! Your manuscript is now with our production department. 

Kind regards, 

on behalf of

Dr. Giuseppe Vittorio De Socio 

Academic Editor

PLOS ONE